# Discovering and Mapping Colloquial Terminologies Describing Underutilized and Neglected Food Crops—A Comprehensive Review

**DOI:** 10.3390/foods12122428

**Published:** 2023-06-20

**Authors:** Szymon Wojciech Lara, Amalia Tsiami, Peter Cross

**Affiliations:** London Geller College of Hospitality and Tourism, University of West London, St Mary’s Road, Ealing, London W5 5RF, UK; szymon.lara@uwl.ac.uk (S.W.L.); peter.cross@uwl.ac.uk (P.C.)

**Keywords:** underutilized food crops, heirloom, landrace, heritage, orphan crops, ancient crops

## Abstract

Global levels of biodiversity and dietary diversity are decreasing, leading to food and nutrition insecurity. This is partially due to the homogenization of the global food supply with commodity crops. The reintroduction or introduction of neglected and underutilized species, minor, forgotten, and indigenous crops and landrace varieties to the wider food systems and further diversification have been outlined as the future strategies for tackling the above by the United Nations and the Food and Agriculture Organization in their policy frameworks. Most of the above species/crops are marginalized and only used across local food systems and in research. With over 15,000 different seed banks and repositories worldwide, information transparency and communication are crucial for database searching and their effective utilization. Much confusion persists around the true nature of those plants, and this prohibits the efficient utilization of their economic potential. A linguistic corpus search and a systematic literature review were conducted using the six most popular collocates to the above terms, which were as follows: ancient, heirloom, heritage, traditional, orphan, and the more distinct term ‘landrace’. The results were interpreted using the Critical Discourse Analysis method. The definitions’ findings show that heirloom, heritage, and ancient are mainly used in the United Kingdom and USA, where they are used to describe ‘naturalized’ and ‘indigenized’ or ‘indigenous’ food crops with a strong affiliation to ‘family’ and the ‘act of passing seeds down from generation to generation’. Orphan crops, on the other hand, are often described as being ‘overlooked’ by growers and ‘underfunded’ by researchers. Landrace is most strongly affiliated with ‘locality’, ‘biocultural diversity’, and ‘indigenous’, and with genomics literature, where the characteristics are often discussed in the context of genetics and population biology. Contextualizing, most of the terms were found to be ‘arbitrary’ and ‘undefinable’ due to their continuing evolution in the socially accepted form of language, perhaps apart from landrace. The review has retrieved 58 definitions for the mentioned 6 terms, together with the primary key terms creating a tool to facilitate a better inter-sector communication and aid in policy.

## 1. Introduction

According to the World State of Plant Fungi Report, the global levels of biodiversity are decreasing at a worrying rate, where two in five plants are threatened with extinction [1]. The agricultural sector, together with aquaculture, are the major contributors to the loss of biodiversity, with 32.8% combined impact to extinction threats [1,2,3]. This is partially due to high-input intensive agricultural practices, based on exhaustive methods, such as monocropping, which are grounded in the “cheaper food” design paradigm [4]. This paradigm has been the primary motivator for the unsustainable practices across the entirety of food systems, making populations vulnerable to future food and nutrition insecurity, especially in the eve of the looming Climate Crisis, leading to poor resilience of production systems and livelihoods [4,5]. 

Despite the negative outcomes, these agricultural methods are still crucial for maintaining a sufficient and stable production of food, in both developed and underdeveloped countries [6]. There are two major factors contributing to this: predictability and price. It is argued that the drastic elimination of these practices from global food systems is unfeasible and would lead to further problems, such as hunger and poverty, and contribute to a decrease in dietary diversity [1,2,3,4]. Simultaneously, researchers are warning of the prolonged consequences of a lack of action, where future populations could also be put at risk of the above issues [7,8,9]. It is, therefore, necessary to consider different strategies for a sustainable change in food systems, where agrobiodiversity-oriented farming methods can be introduced feasibly, without damage to existing food provenance guarantees, with considerations for efficiency, sustainability, nutritional quality, dietary diversity, and economic prosperity [7,8,9]. 

The Food and Agriculture Organization (FAO) and the United Nations (UN) have stated that the process of diversification of agricultural practices across food systems, amongst others, is a good strategy for tackling the induced food and nutrition insecurity. The process of crop diversification has been linked to a decrease in the aspects of “low food diversity”, “monoculture”, and “genetic erosion” and correlated with an improvement in the overall “sustainability” of food production, consumption, nutritional quality, and dietary diversity [7,10,11,12,13,14,15,16].

Crop commodity diversification also corresponds with an increased resistance to other threats, such as plant disease, climate change, and soil depletion [7,14,15,16]. Furthermore, the importance of minor crops within cultural and food heritage paradigms, and the major role of this heritage in their long-term conservation, are also significant. As explained by Ryan et al. [17], some minor crops, which tend to be of high local importance, have their own stories and “timing of change, being additionally connected to transitions in the ways they are processed and transformed into foods and to shifting food preferences”. 

A key characteristic of many minor crops lies in their ongoing popularity within local food systems, as most can be classed as underutilized, neglected, forgotten, or minor in the grand scheme of regionalized and globalized agriculture. In other contexts where agri-development has already displaced the local varieties, some rural communities deliberately pursue low-input farming with neglected and underutilized species (NUS) and landrace varieties of crops. However, the overall tendency to lean towards the homogenization of the food systems, as well as at the local and indigenous levels, is progressive, leading to varietal and genetic loss [18].

Seed banks act as the major institutions for preserving crop seeds and associated information collection; however, this is often limited to the English/Latin colloquial and taxonomic species names, which can include agri-developed cultivars and landrace (LR) species, which means that it can be difficult to distinguish the two. Moreover, the cultural context of LRs [19,20] is ‘culturally/people’-based, as opposed to cultivars strictly referring to ‘true to type’ propagation, which is often commercially driven. These parameters, together with accession year, and sometimes with unspecified collection location, make their popularization—that could otherwise draw on perceptions of heritage and local environment suitability—challenging. This scenario has developed, especially as seedbanks’ primary purpose is seed/genetic preservation and crop development, not revival [19,20]. 

In both the scientific and non-scientific literature, these crops are often described with the following overarching English terms: neglected, underutilized, minor, forgotten, and indigenous, which are frequently seen as interchangeable and are very often used to describe “uncommercial” and “wild edibles” or “semi-domesticated” varieties of crops, and sometimes localized and cultural cultivars of commodity crops or landraces [7,13,21,22].

Furthermore, the literature often use NUS to describe neglected and underutilized crops, as well as wild species that have potential for domestication or commercial cultivation [23,24], as opposed to minor crops and forgotten crops, which refer specifically to domesticated crops and sometimes specific cultivars and are used interchangeably with other terms, namely the following: (1) orphan, (2) heirloom, (3) heritage, (4) ancient, (5) traditional, and sometimes (6) landrace crops. A common trend is observed with the above terms being used solely in a descriptive format and coexisting in the scientific literature as the direct collocates of NUS, minor, and forgotten, and vice versa [1,7,13,21,22,23,24,25,26,27]. Landrace is a unique term in this debate, as it is used colloquially alongside the other terminology relating to neglected crops; however, landrace definitions are based more closely on a botanical framework, as discussed further below [27]. The additional terms (noted above 1–6) can also be found across a range of online sources, such as blogs, community websites and gazettes, seed bank databases, and other grey literature, and are adapted by gardeners, allotment holders, and crop ‘seed guardians’ across the English-speaking world [25,26,27].

The successful utilization of these crops for the fight against food and nutrition insecurity depends greatly on their visibility to potential growers, researchers, policy makers and investors across the value chains and markets. A foundational challenge lies in the genre of definitions, or their absence.

The understanding of the meanings embodied within these terms varies greatly across the literature, sometimes leading to problems in intersectoral communications, for example, between the researchers, the seed banks, and the growers [28]. Across the literature from multiple disciplines, these terms are used in various contexts and their usage is predetermined to subjective variables, such as the scientific discipline, the language, and the form, such as in food science or product development, where many so-called ‘novel foods/ingredients’ are in fact common across other sectors of the food systems, sometimes being commodity-like in one region of the world but completely neglected in other.

The terms are rarely defined, and most often one-word lexemes are used, in the form of connotations assigned as synonyms to provide a description and not a definition [25].

Villa et al. [27] shows information on the etymology of the term *landrace*, together with a review of pre-existing definitions and descriptions, including popular synonyms, such as ‘farmers varieties’, highlighting the sophisticated nature of the word and that it may never be suitable for a strict definition as the characteristics of the hypothetical *landrace crop* have also not been well-defined. Landrace definitions also refer to genetic aspects and population biology, and there are differences in how the terminology may fit different groups of plants [27]. A deeper lack of strict definitions and boundaries is present for the remaining five terminologies, as those are used interchangeably and are more categorial.

Based on the existing meanings and descriptions of the above terminologies, we aim to assess the similarities and differences in which these words are received across the scientific and grey literature, and to present the meaning in an accessible tabular format in order to form a descriptive tool for future researchers encountering terminology issues, limiting the uniformity and, therefore, the transparency of data across the fields.

## 2. Materials and Methods

The search strategy consisted of a primary literature review using the following initially predetermined terms: *neglected*, *underutilized*, *minor*, *forgotten,* and *indigenous* to showcase the language inventory used for such edible plants. The identified key words were (1) *orphan*, (2) *landrace*, (3) *heirloom*, (4) *heritage*, (5) *ancient*, and (6) *traditional*. These 6 terms represent entries into different terminology groupings and were, therefore, considered to be key for the scoping review, as they had the potential to be developed into separate categories, based on embodied meanings from the published literature.

The data search was conducted in 2 stages, firstly through a systematic literature review and secondly through a linguistic corpus search. The conceptual framework developed for this review study was based on the work of Jabareen [29], titled “Building a Conceptual Framework: Philosophy, Definitions, and Procedure”.

### 2.1. Stage 1—Systematic Literature Review

The review was conducted in accordance with the Joanna Briggs Institute’s guidance for Systematic Reviews for Effective Data Synthesis (Joanna Briggs Institute, 2022), with the steps defined below. The literature search was conducted electronically on the following academic databases: (1) Science Direct, (2) Emerald Insight, (3) ProQuest, (4) PubMed, and (5) Google Scholar. An additional search was conducted using an electronic institutional database/library called Summon Search Engine, with access provided on behalf of the University of West London [30]. A total of 2 separate searches were conducted across each database using 2 Boolean Codes with the 6 key words embodied within. These were as follows: ((*landrace crop definition*) OR (*orphan crop definition*) OR (*heritage crop definition*) OR (*heirloom crop definition*) OR (*ancient crop definition*)) OR (*traditional crop definition*)) and: *landrace definition* OR *orphan definition* OR *heritage definition* OR *heirloom definition* OR *ancient definition* OR *traditional definition.*

#### Inclusion and Exclusion Criteria

The searches were limited to peer-reviewed and published journal articles only, although some non-peer-reviewed material was also considered for inclusion. Grey literature was not considered for inclusion at this stage. The searches were not limited by any date restraints, the geographical location of the published articles, or the area of study, as those criteria did not seem relevant and could have led to discrimination against valuable articles. The articles were required to be written in English, with the limitation of this highlighted throughout this review article. In order to be included in the review, the article had to show significant correlation to the topic under investigation, with the key words mentioned somewhere throughout the text.

### 2.2. Stage 2—LinguisticCorpora Search

In addition to the scoping review, we also ran online searches using the British National Corpus (BNC). The BNC is a renowned tool, frequently used by researchers from the field of English language and linguistics to determine the frequency of occurrence of specific terms and their contexts—“BNC is a 100-million-word collection of samples of written and spoken language from a wide range of sources, designed to represent a wide cross-section of British English from the later part of the 20th century, both spoken and written” [31,32,33]. The selected 6 terms were individually inserted into the search engine on the BNC and run to automatically generate findings. The database also created a detailed analysis of the findings, including the major themes, which were grouped according to the nature of resources available. Further to that, the BNC also showed the major (usually top 10) synonyms, constructs, and other language features relating to the inserted key word. The findings generated through this search method were used as support for the main findings retrieved through the scoping review. The inclusion criteria for these findings were based on the relevance to the topic, meaning that constructs with no straight correlation to the area under investigation (food systems, food, and nutrition security) were excluded from the final findings, for example: *art theme.* Some relevance to the mentioned topics had to be present, for example: *botany theme*, *agriculture theme*, and *gardening theme*. The key constructs/definitions from both searches (Table 1) were combined and are presented in Table 2. Furthermore, for the term *landrace*, an additional resource was used, the American English National Corpus. This was due to lack of entries available on the BNC, and it has been acknowledged to be equivalent to the BNC.

### 2.3. Shortlisting, Screening, and Selection of Evidence

Through the scoping review, we identified a total of 517 records using the Boolean codes for all databases combined. The records were imported into a software program called Zotero 6.0.16, where the duplicates were removed, resulting in 496 records (see the PRISMA flow chart in Figure 1).

Title and abstract screening of the records was then undertaken. We searched for the previously mentioned 6 key words, and articles showing some relevance to the review were considered for further inclusion. Other articles that did not contain relevant key words but showed some indirect relevance to the area of study were considered for further screening. The removal of irrelevant articles resulted in 53 records. A full-text screening of these records was then undertaken, where, in addition to the previously mentioned key words, we also looked at the connectors used in the definitions across the literature, which were as follows: “*is*”, “*may*”, “*can*”—*be defined* “*as*” “*defined*”, and “*described*”.

### 2.4. Data Classification and Analysis

For the findings from the Stage 1 scoping reviews, the shortlisted articles were analyzed using the NVivo 12 Pro software program. Highlighted fragments of text were divided into the following 2 sections: (1) *intentional definitions* and (2) *ostensive definitions*, meaning complete definitions and descriptors, respectively. In the instance of a definition of one of the 6 terminologies, the relevant fragment of the article was selected and imported into the *intentional definitions* section of results. In order for the text to be classed as an intentional definition, it had to (1) express the overall meaning of the term and (2) be of a determinant nature to the term. All other relevant sections of the texts were classed as ostensive definitions, as the meaning of the term was usually (1) touched upon briefly in a secondary importance and supported with existing examples, more in a (2) description-like format, often for the purpose of providing some background information in the answer to an overarching question.

The data from both stages were analyzed using the Critical Discourse Analysis (CDA) method by Fairclough [33]. CDA was chosen as the appropriate theoretical framework for this research as different texts use linguistic strategies to reflect their ideological positions and CDA decodes these ideologies to show the power structures constructed in the discourse [52]. We analyzed the findings at the textual dimension to critically understand the choices and patterns in vocabulary, such as wording and chosen metaphors, to investigate the ideologies, the grammar, cohesion, and text structure for each of the key words.

### 2.5. Data Presentation

The data from both stages are presented and discussed according to Fairclough’s [33] three dimensions of CDA. Fairclough sees discourse as being three dimensional, including the textual dimension, the discourse practice dimension, and the social practical dimension.

The selected fragments of text from Stage 1 are quoted and discussed simultaneously. Whereas the words identified through the BNC for Stage 2 were thematically analyzed and discussed. It is important to note that identifying encoded ideologies is an artificial process and the found ideologies can be separated imprecisely in different ways, as they are all intertwined. The analysis was conducted on the 6 terms and was organized alphabetically, in separate stages. The total findings from both stages were presented tabularly.

## 3. Results

This section shows the overall results from both the qualitative (scoping review) and the quantitative (BNC) studies. The systematic scoping review approached a yield of 517 records. Out of those, only 33 were shortlisted for final data extraction, as shown in the PRISMA flow chart in Figure 1. The citations of the identified articles (19 out of 33) are presented in Table 1, as well as the terminology of the crop discussed in each publication. The dominant (key) terms were selected, and those were orphan, traditional, ancient, heirloom, heritage, and landrace. The definitions have been listed according to their frequency of use, from the most common to the least common. The green boxes indicate the definition occurrence of the selected key term, with the presented definition somewhere in the retrieved literature or in the BNC, as presented in Table 2. A full list of the shortlisted articles is available in Appendix A.

The sections below show the qualitative and quantitative data retrieved through the systematic review of the literature and the BNC search, respectively. Each key word is presented and discussed separately. The results below show the intentional definitions and the BNC lexeme for the selected key words, presented in qualitative and quantitative formats.

### 3.1. Term—Ancient

The word ‘ancient’ has been loosely defined as “*of or from a long time ago, having lasted for a very long time*”, and this construct is the dominant definition of the word [53]. The findings indicated that *ancient* seems to be often used to promote alternative grains for flour production, baking, and can be found on some restaurant menus. Furthermore, we have identified two articles (see Table 1 for no [41,42]) that provided intentional definitions and zero ostensive definitions for the term. The focus was drawn onto the aspect of historical significance through words such as *ancestors* and *generations*. There are some similarities to *heirloom*, which also has been defined through a similar ‘historical/family’ lens, but more as *interpersonal* and *intergenerational* relations, as follows:

“*Ancient grains are represented by populations of primitive grains, which were not subject to any modern breeding or selection, and which retained characters of wild ancestors, such as large individual variability, ear height, brittle rachis, and low harvest index*”—[41]

“*Ancient cultivars are those that have been passed down for the generations without alteration*”—[42]

In addition to the definition search, we conducted a corpus search on the BNC database. There were 4846 entries across the entire database, and the analysis revealed that the dominant set of topics was from the field of *ancient history* and *archaeology,* with the 3 most popular collocates being: (1) *history*, (2) *culture,* and (3) *tradition*. Overall, the analysis showed that the topics, collocates, synonyms, and clusters did not resemble any significant correlation between the word *ancient* and any relevant disciplines related to food systems and food and nutrition security. On the other hand, the analysis of the ostensive definitions found that the most popular synonyms were (1) *traditional*, (2) *historical,* and (3) *heritage,* which were found in clusters describing the importance of *ancient farming* and *cultivars*. Based on these outcomes, the existing definitions of ancient are sufficient in meaning; however, the word can perhaps be drawn towards the aspect of edible grains, especially when derived from the grey literature. This may lead to some possible confusion amongst the general public, as many so-called ‘ancient grains’, such as spelt, are often found to be domesticated and popularized significantly later than wheat or barley, which are all considered as more ‘*modern*’ [53]. Therefore, the term ‘ancient’ reflects more on older landrace/orphan crops versus hybrids and other improved cultivars.

### 3.2. Term—Landrace

According to the Cambridge Dictionary [53], the dominant definition for ‘landrace’ is as follows: “*a variety (= type) of a crop or a breed of an animal that has developed over time to suit the conditions of a particular local area*”. The word is also widely used in the botanical context and does not seem to have any other contradictory definitions or meanings.

We have identified 44 references in 12 different articles, and 5 of those were classed as intentional definitions (article numbers: [27,35,37,39,49]), and the remaining were classed as ostensive definitions, with the key journal article titled ‘*Landraces: A review of definitions and classifications’* authored by Villa et al. [27]. The document includes various existing definitions of the term from the pre-World War One era, when the term was vaguely used. It was only after the Second World War that it started to be recognizable across food system practitioners [21]. As previously mentioned, Villa et al. [27] discusses the etymology of the term *landrace*, together with pre-existing definitions and descriptions, highlighting the sophisticated nature of the word and that it may never be suitable for a strict definition, as the characteristics of the hypothetical *landrace crop* are difficult to define well because of the complexity of the genetic aspects and the population biology.

The proposed definition presented by Villa et al. [27] is a working definition, as outlined in the article; nevertheless, it is still the most widely accepted version, especially across plant sciences.

“*Dynamic population(s) of a cultivated plant that have historical origin, distinct identity and lacks formal crop improvement, as well as often being genetically diverse, locally adapted and associated with traditional farming systems*”—[27]

The other identified definitions include the following:

“*Each crop landrace has a specific local name assigned to it, highlighting its features and importance to the particular habitat and representing the class of humans inhabiting that area*”—[35,37]

“*An autochthonous landrace is a variety with a high capacity to tolerate biotic and abiotic stress resulting in a high yield stability and an intermediate yield level under a low input agricultural system*”—[49]

Another definition, which was proposed more recently, shows some correlation between the terms *landrace* and *traditional*, and is synonymized with *heirloom.*

“*Landraces–defined as traditional cultivars developed over time after adapting to both natural and cultural environments– or heirloom cultivars*”—[39]

We have found that many journal articles do not provide their own definitions of the term. This could have been caused by the complex nature and continuing evolution of the word. This has been outlined in the Villa et al. [27] publication, in the following way,: “*As landraces have a rather complex nature it is not possible to give an all-embracing definition as it would result in a description*”.

We conducted a search for the term ‘landrace’ using the BNC, however, no entries were found. We then conducted the same search using the AENC, and identified 24 entries; however, analysis was unavailable. The AENC was treated as a secondary data source and only in the instance of lack of information on the BNC. The lack of data on the BNC could indicate the scarcity of this particular term amongst general websites and mean that this word (in its colloquial meaning) is used more widely across the literature in the USA rather than in the UK, hence, it was absent from the BNC. It is important to note that the AENC search could be limited by geographical bias, as certain entries might originate from sources outside of the USA but be written in American English. The term ‘landrace’ seems to be more recent in use across Britain and was likely derived from the USA agricultural sector at some point in the twentieth century [27].

The ostensive definitions showed that the most popular synonyms for the word were as follows: (1) *traditional*, followed by (2) *primitive*, and then (3), *remote, local, medicinal, natural, old, sustainable, conserved, unstandardized, adapted, under-represented, undocumented, misidentified, wild, native,* and *heritage.* Based on these findings, as well as the initial literature review, we have concluded that there is no unitary definition for the term ‘landrace’. Different communities of practice possess different understandings of the meaning of the term. Sometimes, *landraces* are referred to as *heritage varieties,* or vice versa. This often occurs amongst professional chefs who prefer the use of the term *heritage* and in plant sciences [27,54,55]. The issue of misidentification is also common, as sometimes there are no ‘reliable’ tools to define between *landraces* and other cultivars amongst communities of growers, unless the growers save farm seed as opposed to buying commercial cultivars [22,27].

The term ‘landrace’ refers exclusively to cultivated plants, with a focus on the elements of locality, local adaptation, genetic diversity within populations, and the possession of some historical/heritage elements that determine the crop’s uniquity, often not just in botanical contexts but also social contexts. *Landre-crop* is yet another lexeme that is frequently used across the literature and is used equivalently to the other six terms as well as NUS, forgotten, and *minor*, resulting in a slightly different meaning when compared to plain *landrace,* which is more of a definition itself, in an attempt to make the plant categories more botanical (scientific) rather than colloquial.

### 3.3. Term—Heirloom

For the term ’*heirloom’*, we have identified 35 references in 9 various journal articles. Four of these references [36,39,42,43] were classed as intentional definitions and the remaining were classed as ostensive definitions, where the word has been synonymized with *landrace* on three occasions. The dominant dictionary definition for the term is as follows: “a valuable object that has been given by older members of a family to younger members of the same family over many years” and the food-related definition is: “a fruit, plant, or seed of a type that has existed for many years” [53]. The definitions indicate that the determining factor for naming a crop *heirloom* is its historical significance, to pair with the social construct of the family’s heritage. Restaurateurs tend to use this term in menu creation and advertisement, but, interestingly, this is often practiced only in relation to tomato-based dishes [43].

“*Heirloom varieties, commonly defined as having been grown for over 50 years, have been grown and harvested for multiple generations*”—[36]

“*Heirloom cultivars are generally characterized as traditional or older cultivars that are open pollinated, passed down from gardener to gardener or handed down in families, and often not used in large-scale agricultural enterprises. The definition of heirloom varies, and the term does not carry a precise scientific designation*”—[39]

“*The term ‘heirloom’ itself generally applies to varieties that are capable of being pollen-fertilised and that existed before the 1940s, when industrial farming spread dramatically in the USA and the variety of species grown commercially was significantly reduced*”—[43]

“*HRP defines heirloom rice as cultivars that have been ‘handed down for several generations through family members and grown by small landholders in their ancestral farms*”—[42]

“*Heirloom rice as a social construction that facilitates the cooperation between the CHRP and HRP partners and enables the commodification of what was formerly an anti-commodity*”—[42]

The British National Corpus search resulted in 45 entries across 21 topics related to the field of food systems, with the 5 most popular being *seed*, *tomato*, *variety*, *garden,* and *plant*. The collocates were similar, as they included the above words and several additional words such as *family*, *grow*, *inherit*, *organic,* and *open-pollinated.* The clusters were also identical, as they included the above words.

The analysis of the data from the scoping review showed that the most common ostensive definitions involved the following words: *anti-commodity, non-commercial, traditional, culture, family, local, landrace, organic, open-pollinated, wild, regional, non-hybrid,* and *genetically distinct.* It seems that the *open-pollinated* characteristic is a determining factor for the heirloom classification of plants. The identified definitions varied greatly, with some touching on the aspect of the decreasing popularity in naming these crops as *commodities*. The commodity status seems to be replaced with societal relations in the form of family and the “handing down” of crops through generations. In conclusion, the term ‘heirloom’ resembles some relevance to *culture*, *community,* and *the act of food production*, all of which refer to the overall indigenous food system of rural settlements, which can prove to be a valuable source of new ingredients, flavors, and culinary concepts [19,56].

### 3.4. Term—Heritage

Not a single intentional definition for the term ‘heritage’ has been identified through the review. This is likely to be caused by the insignificance and broadness of the term, indicating its ‘common’ usage. Therefore, the meaning of this word is obvious and there is no major need to redefine it. Although, one journal article [48] on *cereals* stated the following: “*in this paper the term (heritage cereals) is meant to include ancient cereals, landraces and older varieties*” [48], but no strict definition has been found. The Cambridge Dictionary possesses a definition of the word, but in a broader context, as follows: “*heritage features belonging to the culture of a particular society, such as traditions, languages, or buildings, that were created in the past and still have historical importance*”; however, no correlation to botany or gastronomy has been identified. Furthermore, it is important to mention that heritage crops, most often tomatoes, are correlated with traditional agriculture and society, similarly to the previously mentioned concept of heirloom crops and are often cited in policy literature [57,58].

On the other hand, the BNC has identified 1937 inserts of the word across the entire database. The 20 most popular topics were of a similar nature to the previously described *ancient,* as these are mainly orientated around the aspects *of historical significance*, *museums*, *monuments*, and *landmarks*. The collocates and synonyms also relate to the above topics, and the word was synonymized with the term *traditional.*

### 3.5. Term—Traditional

We have identified five references across four different journal articles (No: [14,22,25,49]) that provided some ostensive definitions for the term ‘traditional’; however, no intentional definitions were stated (apart from 41 and 47), as with the term ‘heritage’. ‘Traditional’ is also used in an ostensive manner to describe foods that are of certain “*historical value and significance*” [56]. The term ‘traditional’ has been used in a broad form to describe heirloom crops as follows: “*heirloom crop cultivars are traditional cultivars grown for a long time (>50 years)*” [37]. It may seem logical to establish this term as significantly broader in meaning when compared to heritage, and the remaining terminologies are more specific in the meaning, making *traditional* an overarching term that encompasses the remaining five terms.

The BNC search resulted in 9599 entries, with the dominant sources being of an academic nature. The 20 most common topics were of various natures, with the most relevant to the field of study being*: tradition*, *culture*, and *authenticity,* but with no strong correlation to food systems or FNS.

The analysis of the scoping review did not reveal any significant ostensive definitions. Traditional crops were described as “*often low yielding*” by Oldfield and Alcorn [45]; however, some orphan crops tend to have higher yield ratios when compared to their improved relatives [50]. It is important to note that the term *traditional* is frequently used by various organizations, such as the FAO and the UN, but it is less popular amongst some academic disciplines. For example, in anthropology and ethnobotany searches, it is argued that *traditional* resembles ‘static-ness’ and does not reflect the true form of the indigenous food systems, which are believed to be often evolving; therefore, terms such as ‘customary practices’ are used instead [59]. From the gastronomical perspective, the word *traditional* is often used as a descriptor of dishes that have some historical background, often related to ethnic communities or regions.

### 3.6. Term—Orphan

The search resulted in 23 references from 8 different articles [34,38,40,44,46,47,51,60]. We shortlisted four definitions for the term, where some were found multiple times across six various documents. The main concept that can be extracted from the below definitions is the *economic importance*, together with two key words, *underutilized* and *neglected*. It seems that *orphan crops* are perceived as *less known* and *under-researched* with *underexploited value* due to *insignificant investment and academic attention*.

“*So-called orphan crops are underutilized species that have local significance, especially for small-scale farmers, but are neglected on a global scale*”—[40,44]

“*Orphan crops are those which are grown as food, animal feed or other crops of some importance in agriculture, but which have not yet received the investment of research effort or funding required to develop significant public bioinformatics resources. Where an orphan crop is related to a well-characterised model plant species, comparative genomics and bioinformatics can often, though not always, be exploited to assist research and crop improvement*”—[34]

“*What is an orphan crop? In this context, we define it as a plant species which is grown as a food, animal feed or other crop of some importance in agriculture, but which has not received the investment of research effort—or of funding, which often amounts to the same thing—required to develop significant public bioinformatics resources.*”—[34]

“*Orphan crops are comparatively underexploited or underutilised food plants characterized as having relatively low or no perceived economic importance or agricultural significance in advanced economies, meaning they receive relatively little research and development attention*”—[38,46,47]

The BNC search resulted in 153 entries, mainly from *fiction* and *media* sources. The top 20 topics did not resemble any significant correlation to the field of study. The most popular collocates were *adopted*, *widowed, rescued,* and *abandoned*. These words could be interpreted from the botanical perspective, especially when looked at through the lens of the above definitions, where ‘orphan’ is linked to *locality* and *low economic significance*.

The scoping review search identified the following key words: *underutilized*, *unfinanced*, *neglected*, and *native*. Additionally, the following ostensive definitions were also identified:

“*They (orphan crops) are often overlooked by researchers, despite valuable traits that are promising for emerging markets*”—[50]

“*A huge difference exists on the way orphan or underutilized crops received financial investments for research and development*”—[51]

Based on the above results, it can be seen that *orphan* is often used as a reference to *underutilized and neglected crops*, minor, forgotten, and indigenous, including *cultivars*, as well as wild edible plants. This seems to be of significant value when compared to the remaining terminologies, as those have a stronger tendency to lean toward cultivars and are used more in agri-literature.

## 4. Discussion

As outlined by Villa et al. [27], the term ‘landrace’ has been built on from previous definitions—originally developed at the beginning of the 20th century by Von Rünker (1908) and Mansholt (1909).

The defining entries for the term ‘orphan’ have shown a tendency to lean towards the practicality and usability of the hypothetical *orphan* plant. However, it seems that the plants possess promising attributes, but are yet to be utilized due to a lack of funding and research, as explained by Varshney et al. [47]. Sogbohossou et al. [46], and Chiurugwi et al. [38]. Further to that, *orphan* appears to be used interchangeably with NUS or just *underutilized, minor, forgotten, lost,* and *priority crops*, mainly across the botanical sciences and agri-sciences [61]. Whereas, *heirloom* plants seem to be understood as social constructs, where family and community and the non-commercial and traditional food aspects are the determining factors, which are found more often in grey literature. Within *heirloom*, the human elements refer strongly to the act of *passing down,* which could mean that these are human cultivars that are being preserved, but not the wild varieties. This goes along with the existing definitions found in the plant sciences literature [39,42]. Moreover, *heirloom* plants have been referred to as *locality* more often than *orphan.* The term *landrace* seems to be multidisciplinary in the way that it is described. The descriptions span from *locality*, *communities,* and *remoteness* to *medicinal properties* and *sustainability*. There is some correlation between *landrace* and *orphan*, as shown through the *under-represented* factors, as both types of plant seem to possess that characteristic.

All of these foods are also sometimes referred to as *indigenous*, which botanically refers to the crops’ original regions of domestication and diversity. When referring to the identified papers, discussions are oriented on the act of initial crop domestication, planting, harvest, storage, and reformulation, together with cultural aspects such as gastronomy and beliefs. Often, the term ‘indigenous’ refers more specifically to foods within indigenous peoples’ food systems, which can then include both indigenous crops that originated in those parts of the world and other traditionally consumed foods that were introduced in the past [56]. Crops that are grown as landraces still within rural and indigenous food systems can include crops grown for millennia, or else more recent historic introductions, but essentially, they have been grown for long enough to become locally adapted and to have been adopted into the food systems for enough generations to be considered as *traditional* [13,62]. However, it is important to note that *landrace* is often outlined as a separate category in common UK legislature [20,56]. Seedbanks play a crucial role here, functioning as the repositories of seeds and genetic material that can used in research but are not designed to facilitate the actual crop revival into the food systems [20,63,64,65,66].

The meaning of *traditional* is much broader when compared to the remaining five terms. The information embodied in the description for the term was also multidisciplinary and often lacked clarity. This could also be due to the much wider usage of that word in other contexts, as shown through the corpus search, where the number of identified entries on the BNC for *traditional* reached 9599, showing the abundance of that word. There is some overlap with other terms, such as *heritage, ancient,* and *heirloom.* The historical attributes seem to be dominant. *Ancient* is also correlated with *history*, *tradition*, and *culture*, but also encompasses *primitiveness* and *the act of passing something down from generation to generation*, which is also emphasized through *ancestry* or *heirloom*. *Heritage* is narrower, however, in the sense that it most often refers to specific cultivars, especially those of tomatoes and oats, which has been found multiple times throughout the shortlisting process for this review.

Further research should be carried out using other non-English terms that show linkage with some of the identified terms through this systematic review. Moreover, it is worth noting that some terms could have been missed due to the literature publication formats, a lack of digitalization, or document accessibility through modern search engines. A good example of this is the Series of Volumes of Lost Crops of Africa, where the term ‘lost-crops’ is frequently used [64,65,66]. Further English search terms could include indigenous, lost, minor, and forgotten and an analysis of the meanings embodied there could be used in the promotion of forgotten crops along the food supply chains, through their popularization along the value chains, directing investments, markets and focusing attention of policy makers [67,68,69].

## 5. Conclusions

A scoping literature review was conducted on the existing definitions of 6 key terms (*orphan, heirloom, heritage, ancient, traditional,* and *landrace*), according to the Joanna Brigs Institute’s guidance and the PRISMA model, using 33 shortlisted articles. Out of those, 58 definitions have been identified as linking to the above key terms. The key term *landrace* has a botanical and taxonomic framing, and is used to describe crops that are uncommercial, localized, and indigenous. In addition, sometimes *traditional* and *landrace crops* are also determined in genetic and population biology characteristics. *Orphan*, *heirloom,* and *heritage* are used mainly in the UK and the USA and can refer to the minor cultivars and sometimes-forgotten *landraces*. *Orphan* is also commonly used in the agri-development literature globally. *Ancient* generally refers to old varieties/cultivars, species, or wild edibles and cultivated wild plants, and is often used for grains. Some of the above terms can be categorized under *NUS* (neglected and underutilized species), which includes all crops and cultivable wild species that are not major/significant in relation to food systems or NUC (c = crops), which includes all minor crops but not the wild species. *Landrace crops* are a separate category, consisting of both minor and major species. 

Highlighting the ever-changing nature of some of the above terms, alongside a more precise application in legislature, has the potential to improve communication in different disciplines and stakeholders from across food systems. Better framing in both the scientific and non-scientific literature could enhance the transparency and accessibility of knowledge to non-experts in the light of growing popularity of interdisciplinary research. All species, crops, and landraces that are categorized under any of the mentioned terms have the potential to help with the increase in agrobiodiversity, facilitating a better dietary diversity and contributing to the elimination of food and nutrition insecurity caused by many factors, including looming climate change. Future research could focus on the identification and definition of terms from foreign languages. Policy development in the above fields should be open to using a variety of terminologies, including new emerging trends, such as the use of *forgotten crops,* as opposed to prioritizing some terms for all beneficiaries, and consider their complex meanings and linkage to various natural and social aspects, such as geography, language, culture, and development level, in order to increase the level of mutual understanding between researchers, policy makers, growers, and consumers, beyond the English-speaking world. 

## Figures and Tables

**Figure 1 foods-12-02428-f001:**
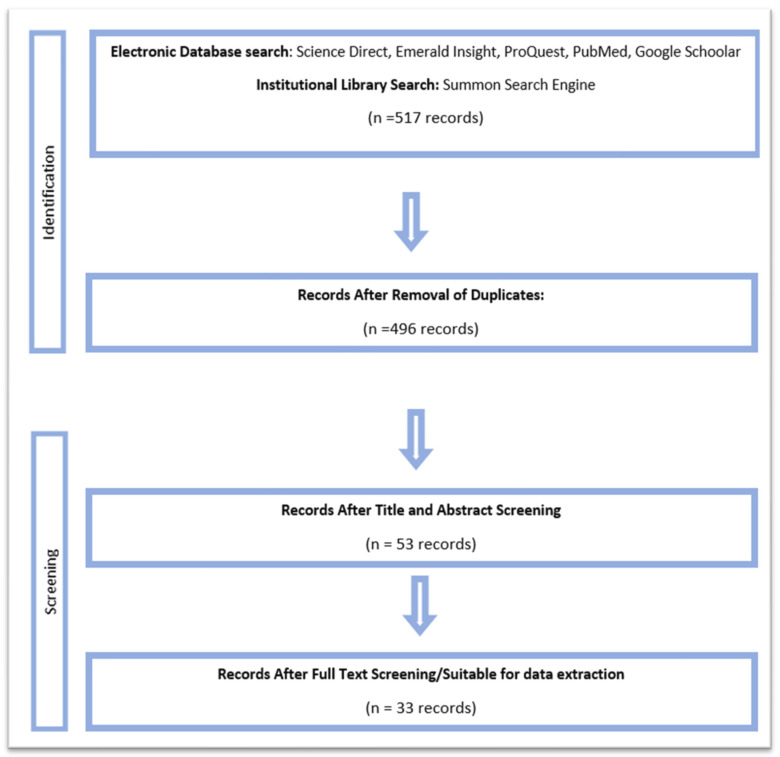
PRISMA flow chart displaying the search and shortlisting processes, as defined in Section 2.3, and the screening process, as defined in Section 2.4.

**Table 1 foods-12-02428-t001:** This table presents the articles retrieved through the scoping review that contain intentional definitions (19 out of 33 shortlisted articles). The remaining articles with ostensive definitions can be found in Appendix A. The article numbers correspond with the chronological occurrence in the text.

No.	Articles with Intentional Definitions Identified Through the Scoping Review (Citations)	Terminology
[27]	(Villa et al., 2005)	Landrace
[34]	(Armstead et al., 2009)	Orphan
[35]	(Benlioğlu and Adak, 2019)	Landrace
[36]	(Brouwer et al., 2016)	Heirloom
[37]	(Casañas et al., 2017)	Landrace
[38]	(Chiurugwi et al., 2018)	Orphan
[39]	(Dwivedi, Goldman, and Ortiz, 2019)	Heirloom Landrace Traditional
[40]	(Epping and Laibach, 2020)	Orphan
[41]	(Giambanelli et al., 2013)	Ancient
[42]	Glover and Stone, 2017	Ancient Heirloom
[43]	(Jordan, J., 2007)	Heirloom
[44]	(Mabhaudhi, et al., 2019)	Orphan
[45]	(Oldfield and Alcorn., 1987)	Traditional
[46]	(Sogbohossou et al., 2018)	Orphan
[47]	(Varshney et al., 2012)	Orphan
[48]	(Wendin et al., 2020)	Heritage
[49]	(Zeven, 1998)	Landrace
[50]	(Tadele, Z., 2009)	Orphan
[51]	(Naylor et al., 2004)	Orphan

**Table 2 foods-12-02428-t002:** In this table, the 58 extracted definitions (most relevant international and ostensive) have been presented together with the 6 key terms. The definitions have been listed according to frequency of use, from the most common to least common, across for the 6 key terms. The green boxes indicate definition occurrence of the selected key term with the presented definition somewhere in the retrieved literature (see Appendix A for a full list of shortlisted articles) or in the BNC/AENC. This table could be used to increase the accuracy of future literature searches and to help differentiate between the key terms.

Key Definitions Identified	Orphan	Traditional	Ancient	Heirloom	Heritage	Landrace
Neglected	Yes	Yes	Yes	Yes	Yes	Yes
Underutilized	Yes	Yes	Yes	Yes	Yes	Yes
Regional	Yes	Yes	Yes	Yes	Yes	Yes
Indigenous	Yes	Yes	No	Yes	Yes	Yes
Tradition	No	Yes	Yes	No	Yes	Yes
Generations	No	Yes	Yes	Yes	Yes	No
Older Cultivars	Yes	No	Yes	Yes	No	Yes
Under-represented	Yes	No	No	Yes	Yes	Yes
Low Yielding	Yes	Yes	Yes	No	No	No
From a long time ago	No	Yes	Yes	No	Yes	No
Culture	No	Yes	Yes	No	Yes	No
Ancestors	No	No	Yes	Yes	Yes	No
Passed down for generations	No	No	Yes	Yes	Yes	No
Traditional Cultivars	No	Yes	No	Yes	No	Yes
Wild	Yes	No	Yes	Yes	No	No
Heritage Tomatoes	No	Yes	No	Yes	Yes	No
Representing the Communities	Yes	Yes	No	No	No	Yes
Natural and Cultural Environments	Yes	No	No	No	Yes	Yes
Remote	Yes	No	No	Yes	No	Yes
Local	Yes	No	No	Yes	No	Yes
Family	No	No	No	Yes	Yes	Yes
Primitive	Yes	Yes	Yes	No	No	No
Small Landholders	Yes	No	No	Yes	No	Yes
Natural	Yes	Yes	No	No	No	Yes
Conserved	Yes	No	No	Yes	No	Yes
Grown for 50 years +	No	Yes	No	Yes	No	No
Historic	No	Yes	No	No	Yes	No
Wild Ancestors	Yes	No	Yes	No	No	No
Genetically Distinct	No	No	No	Yes	No	Yes
Traditional Agriculture	Yes	No	No	No	Yes	No
Adopted	Yes	No	No	No	No	Yes
Variability	No	No	Yes	No	No	Yes
Underexploited	Yes	No	No	No	No	No
Overlooked	Yes	No	No	No	No	No
Unfunded	Yes	No	No	No	No	No
Unresearched	Yes	No	No	No	No	No
Underdeveloped	Yes	No	No	No	No	No
Valuable Traits	Yes	No	No	No	No	No
Promising	Yes	No	No	No	No	No
With Some Importance	Yes	No	No	No	No	No
Authenticity	No	Yes	No	No	No	No
Historic Significance	No	No	Yes	No	No	No
Polen-Fertilized	No	No	No	Yes	No	No
Open-Pollinated	No	No	No	Yes	No	No
Anti-Commodity	No	No	No	Yes	No	No
Organic	No	No	No	Yes	No	No
Non-commercial	No	No	No	Yes	No	No
Non-hybrid	No	No	No	Yes	No	No
Historical Importance	No	No	No	No	Yes	No
Traditional Society	No	No	No	No	Yes	No
Older Varieties	No	No	No	No	Yes	No
Distinct Identity	No	No	No	No	No	Yes
Lacking Importance	No	No	No	No	No	Yes
Tolerates Biotic/Abiotic Stress	No	No	No	No	No	Yes
High Yield Stability	No	No	No	No	No	Yes
Complex Nature	No	No	No	No	No	Yes
Medicinal	No	No	No	No	No	Yes
Sustainable	No	No	No	No	No	Yes

## Data Availability

Data is contained within the article. The data presented in this study are available in the provided tables, references and appendices. Please use the stated article names and DOIs for data searching.

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
