# Peer review of "Discovering and Mapping Colloquial Terminologies Describing Underutilized and Neglected Food Crops—A Comprehensive Review"

_foods, 2023, doi:10.3390/foods12122428_

Round 1

Reviewer 1 Report

This article provides an interesting classification of the different terminology used for NUS crops. It is based on a systematic literature review and critical discourse analysis.

The article would benefit from a better positioning in current food systems analysis, focussing not only on biodiversity but also the importance of dietary diversity. Moreover, attention should be given to the importance of risk management and resilience of production systems and livelihoods.

The analysis is limited to English/US sources, whereas much NUS research is done in local languages and the selection of adequate terms is strongly related to cultural habits and identification.

The scoping phase of the systematic review remains totally unclear. Which are the criteria used for exclusion? Moreover, it is welcome to assess years of publication and countries of origin. 

Differences between 'seeds', 'species', 'varieties' and 'relatives' need to be clearly defined. Some linguistic terms refer implicitly to original crop species, others to varities gebnerated by evolution.

The conclusion is very insatisfactory: it bascally refers to difficyulties and future research.  Authors are encouraged to assess the relevance of the different terms in their specific contexts and to recommend which terms are better suited to particular policy discussions.

Author Response

please find the answers at the attached document

Reviewer 2 Report

Thank you for the opportunity to review the manuscript entitled “Discovering and Mapping Colloquial Terminologies Describing Underutilized and Neglected Food Crops. A Comprehensive Review”, submitted for possible publication to Foods. The research is based on the assumption that the reintroduction of neglected and underutilized crops in food systems could represent a possible solution to guarantee food security and diversify the global food supply chain. The authors have conducted a systematic literature review and have interpretated data using the Critical Disclosure Analysis method. 

I have checked the similarity index with the Compilatio Magister software, and it has been estimated at 12%, which is rather okay and acceptable in the field of similarity and plagiarism. 

First, the abstract does not follow the Instruction for Authors provided by MDPI. Please, delete LL. 30-38. The abstract is clear and comprehensive. It defines (briefly) the theoretical background, the purpose of the research, the methods applied and the main insights. 

The section “Introduction” should provide additional statistics and facts related to the food insecurity, as well as to the loss of biodiversity and to climate change (LL. 43-49). It is important to provide, together with descriptive information, also quantitative data, as to help readers better contextualize the research background and the need to conduct the research. 

The research is confusing. I can see some missing footnotes. For instance, at L. 49 there is “Climate Change 4,5” or at L. 52 there is “countries6”. Where are these footnotes? Why they begin with the number 4 and not with the number 1? Please, revise.  Are they references instead of footnotes? Please, revise. Same for L. 66, etc.

L. 61 it should be “FAO” and not “FOA”. 

L. 80. Although NUS has been cited as an acronym in the abstract, it appears for the first time in the main body of the text. It should be explained its meaning again. 

I suggest the authors increasing the description of the originality and the novelty of the research, by adding some more attracting insights and justifications (LL. 114-116). Also, it should be better clarified the purpose of the research, since at recent is not sufficient. 

As regards “Materials and Methods”, first the research should better identify the literature (and the references) on which the six keywords have been selected (LL. 119-121). The process of identification of the keywords represents the main step to strengthen the scientific soundness of the research. 

The description of the systematic literature review process is not sufficient. Which are the different steps of the Joanna Briggs Institute guidance? Could you please provide some more details related to such an approach? For instance, I also suggest adopting the PRISMA model to conduct systematic literature reviews. The authors should better define the exclusion and inclusion criteria, and also the reasons behind such choices (e.g., why not date restraints?). 

L. 146 “British National Corpus” should be written as follows, as to identify the acronym: “British National Corpus (BNC)”. Further, it should be better identified how the online search has been conducted, and with which purpose. 

L. 169. What are the “Boolean” codes?

L. 171. It appears confusing. Have the authors applied the Joanna Briggs Institute guidance, of the PRISMA guidelines?

The section “Data Classification and Analysis” could be improved further. For instance, it is not clear “why” the articles have been defined into the sections “intentional definitions” and the “ostensive definitions”. Also, it is not clear how the Critical Discourse Analysis works. Could you please provide readers with some more details? For instance, which are the “three dimensions of CDA” (LL. 203-204). It seems that several aspects have been given for granted, whereas they should be clarified. 

LL. 213-214. What does it mean “qualitative” and “quantitative” studies? 

As regards Table 1, it is not clear how the screening stage has been conducted, which has allowed to pass from 496 to 53 articles. 

In Table 2, does “Y” mean “Yes” and “N” mean “No”? Please, clarify. 

The section “Results” and “Discussion” seem rather descriptive and does not clearly provide some empirical/managerial implications, as well as possible suggestions for public authorities and policy makers. The authors are invited to add such insights.

Overall, the English should be spell and grammar checked, and also the Insturction for Authors should be revised. 

Author Response

please find the answers to the document attached

Round 2

Reviewer 2 Report

Thank you for the opportunity to review the manuscript entitled “Discovering and Mapping Colloquial Terminologies Describing Underutilized and Neglected Food Crops. A Comprehensive Review”, submitted for publicatio to Foods. In the light of the reviewer’s suggestions, the authors have revised the manuscript. 

The abstract has been revised with additional details, and also the English has been spell-checked and grammar checked. Also, the abstract clarifies the conclusions. 

The section “Introduction” has been revised by the authors, as to highlight the purpose of their research, as well as the theoretical background, the originality and the nature of “descriptive study”. 

The section “Materials and Methods” is clear and comprehensive. The authors have clarified each stage of the research process, which at current has reached a suitable replicability, and have added references. 

“Results” are clear and descriptive. Also tables are comprehensive. The authors, in addition, have substantially implemented the sections “Discussion” and “Conclusions”. 

Author Response

thank you for the review